# PD-L1 upregulation by IFN-α/γ-mediated Stat1 suppresses anti-HBV T cell response

LanLan Liu[1,2☯], Junwei Hou[1,3☯], Yuxiu Xu[1,3], Lijuan Qin[1,3], Weiwei Liu[1,3], Han Zhang[1,3], Yang Li[1,3], Mi Chen[1,3], Mengmeng Deng[1,3], Bao Zhao[1,3], Jun Hu[1,3], Huaguo Zheng[1,3], Changfei Li[1,3]*, Songdong Meng[1,3]*

1 CAS Key Laboratory of Pathogenic Microbiology and Immunology, Center for Biosafety Mega-Science, Chinese Academy of Sciences (CAS), Institute of Microbiology, Beijing, China, 2 Institutes of Physical Science and Information Technology, Anhui University, Hefei, China, 3 University of Chinese Academy of Sciences, Beijing, China

☯ These authors contributed equally to this work.
* mengsd@im.ac.cn (SM); lichangfei2006@163.com (CL)

**Data Availability Statement:** All relevant data are within the manuscript and its Supporting Information files.

**Funding:** This work was supported by a grant from the Strategic Priority Research Program of the

## Abstract

Programmed death ligand 1 (PD-L1) has been recently shown to be a major obstacle to antiviral immunity by binding to its receptor programmed death 1 (PD-1) on specific IFN-γ producing T cells in chronic hepatitis B. Currently, IFN-α is widely used to treat hepatitis B virus (HBV) infection, but its antiviral effect vary greatly and the mechanism is not totally clear. We found that IFN-α/γ induced a marked increase of PD-L1 expression in hepatocytes. Signal and activators of transcription (Stat1) was then identified as a major transcription factor involved in IFN-α/γ-mediated PD-L1 elevation both *in vitro* and in mice. Blockage of the PD-L1/PD-1 interaction by a specific mAb greatly enhanced HBV-specific T cell activity by the gp96 adjuvanted therapeutic vaccine, and promoted HBV clearance in HBV transgenic mice. Our results demonstrate the IFN-α/γ-Stat1-PD-L1 axis plays an important role in mediating T cell hyporesponsiveness and inactivating liver-infiltrating T cells in the hepatic microenvironment. These data raise further potential interest in enhancing the anti-HBV efficacy of IFN-α and therapeutic vaccines.

## Introduction

Up to 350 million individuals worldwide are currently chronically infected with hepatitis B virus (HBV). Ultimate HBV clearance requires the coordination of the potent T cell immune response and effective humoral immunity. However, HBV-specific T cell response, which plays a vital role in HBV clearance, is severely impaired in chronic hepatitis B (CHB) patients, leading to long-term immune tolerance [1, 2]. Several mechanisms may contribute to HBV-specific T cell exhaustion, including upregulation of co-inhibitory molecules such as programmed death 1 (PD-1), T-cell immunoglobulin and mucin domain-containing molecule 3 (TIM-3), T-cell immunoglobulin and ITIM domain (TIGIT), lymphocyte-activation gene 3 (LAG3), immunosuppressive prostaglandin E2 (PGE2) receptors, cytotoxic T-lymphocyte antigen 4 (CTLA-4), and proapoptotic protein Bcl2-interacting mediator (Bim) on HBV-

Chinese Academy of Sciences (XDB29040000), a grant from One Belt and One Road International Science and Technology Cooperation of Chinese Academy of Sciences (153211KYSB20170001), the Industrial innovation team grant from Foshan Industrial Technology Research Institute, Chinese Academy of Sciences, and grants from the National Natural Science Foundation of China (81761128002, 81621091, 81871297, 81672815, 81471960,81903142) to SM and BZ. The authors acknowledge materials support from Beijing Combio Company and Beijing Tiantan Biological Products Company. The funders had no role in study design, data collection and analysis, decision to publish, or preparation of the manuscript.

**Competing interests:** The authors acknowledge materials support from Beijing Combio Company and Beijing Tiantan Biological Products Company. This does not alter our adherence to PLOS ONE policies on sharing data and materials.

specific CD8[+] T cells, as well as on CD4+ T cells and NK cells [3–5]. Additionally, regulatory T cells and suppressive cytokines also contribute to virus-specific T cell failure [6].

Among the co-expressed inhibitory receptors on T cells, programmed death ligand 1 (PD-L1) plays a critical role in impaired T cell immune responses. Of note, its ligand PD-L1, a 40 kDa transmembrane protein, is constitutively expressed on liver DCs, Kupffer cells, stellate cells, liver sinusoidal endothelial cells, and hepatocytes. Binding of PD-L1 to PD-1 leads to T cell dysfunction by inhibiting T cell activation, causing T cell exhaustion, anergy, and T cell apoptosis, as well as by inducing Treg differentiation [7–11]. In addition, elevated PD-L1 levels in liver were observed in chronic necroinflammatory liver diseases and autoimmune hepatitis [12, 13]. These indicate the immune regulatory function of the liver microenvironment that may lead to T cell exhaustion.

As an first-line treatment option, IFN-α-based therapies achieve a sustained off-treatment response and a more likely functional cure, and prevent occurrence of hepatocellular carcinoma in patients with CHB [14, 15]. Virus-specific IFN-γ secreting CD8+ and CD4+ T cells are believed to play a key role on HBV clearance and control [16–18]. However, both type I/II interferons were shown to promote PD-L1 expression in hepatocytes, which may induce T cell apoptosis [19–21]. Therefore, further elucidating the mechanism of hepatic PD-L1 expression induced by IFN-α/γ and its role in T cell response will shed light on the underlying mechanism of antiviral T cell exhaustion and the unique immunological properties of liver.

Here, we aimed to explore the mechanism of PD-L1 upregulation in hepatocytes by IFN-α/γ and the potential role of PD-L1 in regulating virus-specific T cell responses in liver. The results could provide valuable insights into the modulation of hepatic PD-L1 expression by type I/II interferons, and offer novel therapeutic combination strategies for reversing T cell immune exhaustion in CHB.

## Materials and methods

### Cell lines

The human hepatic cell line L02 originated from normal human liver tissue immortalized by stable transfection with the human telomerase reverse transcriptase (hTERT) gene [22, 23]. The L02 and Huh7 cell lines were obtained from the Cell Bank of the Chinese Academy of Sciences (Shanghai, China) and maintained in the lab. The L02 and Huh7 cell lines were cultured in Dulbecco's modified Eagle's medium (DMEM) containing 10% heat-inactivated fetal bovine serum, 1 g/L of glucose, 1 mmol/L of glutamine, 100 U/mL of penicillin, and 100 μg/mL of streptomycin, and incubated in 5% CO2 at 37°C.

### Plasmids, antibodies, and reagents

The Stat1 expression plasmid pCMV-Stat1, pGL3-PD-L1 promoter-luciferase (PD-L1-wt) and pGL3-PD-L1 promoter-mutant-luciferase (PD-L1-mut) with mutated Stat1 binding site were constructed by our lab. Rabbit Stat1 antibody and phospho-Stat1 monoclonal antibodies were purchased from Cell Signaling Technology (MA, USA). The anti-PD-1 monoclonal antibody (mAb) was kindly provided by Beijing Combio Company (Beijing, China). The PD-L1 monoclonal antibody was obtained from eBioscience (MA, USA). The specific Stat1 inhibitor fludarabine was from Selleck Chemicals (TX, USA). The Dual-Glo® Luciferase Assay System was purchased from Promega Corporation (WI, USA). The human IFN-α and IFN-γ proteins, as well as the murine IFN-α protein were purchased from Sino Biological Inc (Beijing, China). The HBs protein was kindly given by Beijing Tiantan Biological Products Company (Beijing, China). The gp96 and HBc proteins were expressed and purified in our lab respectively as described previously [24, 25]. The recombinant murine IFN-γ protein was purchased from

PeproTech Inc. (NJ, USA). Mouse IFN-γ precoated ELISPOT kit was provided by Dakewe Inc. (Shenzhen, China). The HBsAg and HBeAg test kits were purchased from Shanghai Kehua Bio-Engineering Ltd. (Shanghai, China). HBV nucleic acid amplification fluorescent quantitative assay kit and Alanine Transaminase Assay kit were purchased from Beijing Biodee diagnostics Ltd. (Beijing, China). The sequences of PD-L1 promoter and Stat1-specific siRNA are listed in Table 1.

Western blotting, flow cytometry analysis of cell membrane PD-L1 levels, immunohistochemistry (IHC) analysis, and real-time PCR were performed as described previously [26]. The primers used in real-time PCR are indicated in Table 2. Luciferase reporter assays were performed as described previously [27]. Flow cytometry and intracellular cytokine staining of CD4$^+$ or CD8$^+$ T cells, IFN–γ ELISPOT, serum ALT detection, and virology assessment (serum HBs and HBe Ag, and HBV DNA copies) were performed as described previously [24].

## IFN-α and IFN–γ treatment in mice

Six-week-old male BALB/c mice were purchased from Vital River Laboratories. Six-week-old male BALB/c HBV transgenic mice were purchased from Transgenic Engineering Lab, Infectious Disease Center, Guangzhou, China. Mice were randomly divided into 5 groups (n = 5/ group) and injected with PBS, IFN-α ($5 \times 10^4$ U/kg), IFN–γ ($1.6 \times 10^4$ U/kg), and/or the specific Stat1 inhibitor fludarabine (40 mg/kg) every 3 days for 5 times, respectively. Three days after the last injection, all the mice were sacrificed, and the mouse livers were fixed in formalin and prepared for IHC analysis.

## Combined therapy with anti-PD-1 mAb and HBV therapeutic vaccine

Six-week-old male BALB/c HBV transgenic mice were purchased from Transgenic Engineering Lab, Infectious Disease Center, Guangzhou, China. The HBV transgenic mice were generated with a viral DNA construct, pHBV1.3, containing 1.3 copies of the HBV genome. All transgenic mice were tested positive for serum HBsAg and viral DNA, as well as HBc expression in hepatocytes in their livers. Mice were randomly divided into 4 groups (n = 5/group). Mice were subcutaneously immunized with HBV therapeutic vaccine (10 μg HBs+10 μg HBc + 25 μg gp96/mouse) at weeks 1, 2, and 4, respectively, and/or intraperitoneally injected with anti-PD-1 mAb (100 μg/mouse) at weeks 1, 2, 3, and 4, respectively. Mice were sacrificed at week 9 for antiviral T cell analysis and virology assessment.

## Statistical analysis

All data were presented as mean ± SD, and significance was determined by two-tailed Student's t test unless specified. A P value of less than 0.05 was considered statistically significant. In figures $^*$ indicates for P<0.05, $^{**}$ for P<0.01 and $^{***}$for P<0.001.

## Study approval

Animal studies were approved by the Institute of Microbiology, Chinese Academy of Sciences of Research Ethics Committee (permit number PZIMCAS2011001). All animal experiments were performed in strict accordance with institutional guidelines on the handling of laboratory animals.

**Table 1. This is the sequences of promoter, siRNA.**

| Gene | promoter /siRNA |
|---|---|
| PD-L1 promoter | 5'-TCATAACCAATGCAAGGGCTATCTCAATATTCATT CATTATGCAGTATTTTGAACTGCAGTTGAAATGA ATAAGAAGGAAAGGCAAACAACGAAGAGTCCAAT TTCTCAATTTAGAAAAAGAGAAAAAAAAGAAAAGG GAGCACACAGGCACGGTGGCTCAAGCCTGTAATAT CAGCACTTTGGCGGATCACTTGAGGTCAAGGAGTT CGAGAAAAGAGAGCACCTAGAAGTTCAGCGCGGGAT AATACTTAAGTAAATTATGACACCATCGTCTGTCATC TTGGGCCCATTCACTAACCCAAAGCTTTCAAAAGGGC TTTCTTAACCCTCACCTAGAATAGGCTTCCGCAGCCTT AATCCTTAGGGTGGCAGAATATCAGGGACCCTGAGCAT TCTTAAAAGATGTAGCTCGGGATGGGAAGTTCTTTTAA TGACAAAGCAAATGAAGTTTCATTATGTCGAGGAACTT TGAGGAAGTCACAGAATCCACGATTTAAAAATATATTTC CTATTATACACCCATACACACACACACACACCTACTTTCT AGAATAAAAACCAAAGCCATATGGGTCTGCTGCTGACTT TTTATATGTTGTAGAGTTATATCAAGTTATGTCAAGATG TTCAGTCACCTTGAAGAGGCTTTTATCAGAAAGGGGGAC GCCTTTCTGATAAAGGTTAAGGGGTAACCTTAAGCTCTT ACCCCTCTGAAGGTAAAATCAAGGTGCGTTCAGATGTTG GCTTGTTGTAAATTTCTTTTTTTATTAATAACATACTAAA TGTGGATTTGCTTTAATCTTCGAAACTCTTCCCGGTGAAA ATCTCATTTACAAGAAAACTGGACTGACATGTTTCACTTTC TGTTTCATTTCTATACACAGCTTTATTCCTAGGACACCAAC ACTAGATACCTAAACTGAAAGCTTCCGCCGATTTCACCGAA GGTCAGGAAAGTCCAACGCCCGGCAAACTGGATTTGCTGC CTTGGGCAGAGGTGGGCGGGACCCCGCCTCCGGGCCTGGC GCAACGCTGAGCAGCTGGCGCGTCCCGCGCGGCCCCAGTT CTGCGCAGCTTCCCGAGGCTCCGCACCAGCCGCGCTTCTGT CCGCCTGCAGGTAGGGAGCGTTGTTCCTCCGCGGGTGCCCA CGGCCCAGTATCTCTGGCTAGCTCGCTGGGCACTTTAGGAC GGAGGGTCTCTACACCCTTTCTTTGGGATGGAGAGAGGAG AAGGGAAAGGGAACGCGAT-3'. |
| Stat1-specific siRNA Control-siRNA | 5'-GGGCAUCAUGCAUCUUACU-3' 5'-UUCUCCGAACGUGUCACGUTT-3' |

**Table 2. Real-time PCR primers sequences.**

| Gene | primers sequences |
|---|---|
| Stat1 | Forward: 5'-CAGCTTGACTCAAAATTCCTGGA-3' |
| | Reverse: 5'-TGAAGATTACGCTTGCTTTTCCT-3' |
| NF1 | Forward: 5'-CGAATCATCACCAATTCCGCA-3' |
| | Reverse: 5'-CCACAACCTTGCACTGCTTTAT-3' |
| Stat3 | Forward: 5'-ATCACGCCTTCTACAGACTGC-3' |
| | Reverse: 5'-CATCCTGGAGATTCTCTACCACT -3' |
| PAX2 | Forward: 5'-TCAAGTCGAGTCTATCTGCATCC-3' |
| | Reverse: 5'-CATGTCACGACCAGTCACAAC-3' |
| IRF1 | Forward: 5'-GCAGCTACACAGTTCCAGG-3' |
| | Reverse: 5'-GTCCTCAGGTAATTTCCCTTCCT -3' |
| Stat4 | Forward: 5'-TGTTGGCCCAATGGATTGAAA-3' |
| | Reverse: 5'-GGAAACACGACCTAACTGTTCAT-3' |
| PD-L1 | Forward: 5'-GCTGCACTAATTGTCTATTGGGA-3' |
| | Reverse: 5'-AATTCGCTTGTAGTCGGCACC-3' |
| Actin | Forward: 5'-CATGTACGTTGCTATCCAGGC-3' |
| | Reverse: 5'-CTCCTTAATGTCACGCACGAT-3' |

## Results

### IFN-α/γ induce PD-L1 expression in hepatocytes

We first tested whether IFN-α and IFN-γ could affect PD-L1 expression in hepatocytes. As shown in Fig 1A, PD-L1 expression was increased by IFN-γ in a dose-dependent manner from 10 to 800 U/ml 48 h after treatment. Similar results were observed for IFN-α (Fig 1B). In vivo experiment, BALB/c or BALB/c HBV transgenic mice were intraperitoneally injected with IFN-α or IFN-γ, and PD-L1 expression in liver tissues were examined by IHC. Treatment with IFN-α or IFN–γ induced abrupt increases of PD-L1 levels in both HBV transgenic mice and BALB/c mice (Fig 1C).

We then determine if IFN-α/γ upregulate the expression of PD-L1 by affect its transcription level. Real-time PCR analysis showed that IFN-γ pronouncedly increased PD-L1 mRNA levels (Fig 1D). Similar results were observed for IFN-α (Fig 1E).

### IFN-α/γ induce PD-L1 expression mainly through activation of Stat1

The PD-L1 promoter sequence was subjected to bioinformatics analysis (http://gpminer.mbc.nctu.edu.tw/), revealing that there are several putative transcription factor binding sites in the promoter region, including sites for Stat1, NF1, Stat3, PAX2, IRF1 and Stat4 (Fig 2A). We next tested the effect of IFN-γ or IFN-α on expression of these transcription factors. L02 cells stimulated with IFN-γ for 24 h, and Stat1 mRNA level was the mostly increased (around 10 times) compared to the other transcription factors (Fig 2B). Same results were obtained for IFN-α treatment (Fig 2C). The protein expression of Stat1 and its phosphorylation were also stimulated by IFN-γ or IFN-α (Fig 2D). We mutated the core site CTGAT of Stat1 binding site on the PD-L1 promoter to ACTGC and named it as PD-L1-mut (Fig 2A). As shown in Fig 2E and 2F, Stat1 overexpression by transfection with the plasmid pCMV-Stat1 in L02 cells led to an obvious increase in the wild-type but not mutant PD-L1 promoter activity (~ 3.5-fold) (P < 0.001) and significant elevation of cell membrane PD-L1 levels. PD-L1 promoter activity was increased by Stat1 in a dose dependent manner (Fig 2G). These results indicate that IFN-α/γ upregulates the key PD-L1 transcription factor Stat1.

To further determine the role of Stat1 in IFN-α/γ-induced PD-L1 expression, L02 cells were transfected with PD-L1 luciferase reporter plasmid with a wild type or mutated Stat1 binding site and incubated with IFN-γ for 48 h. As seen in Fig 3A, IFN-γ treatment increased the activity of wild type but not the mutated promoter. Moreover, Stat1 depletion by siRNA or its inhibitor fludarabine largely abolished IFN-γ-induced PD-L1 promoter activation (Fig 3B). Similar results were observed for IFN-α (Fig 3C and 3D).

Similarly, PD-L1 upregulation by IFN-α/γ was mostly abrogated under Stat1 depletion by siRNA or inhibition by fludarabine (Fig 4A and 4B). Inhibition of Stat1 and p-Stat1 by fludarabine was confirmed by western blot (Fig 4C). Similar results were observed in total PD-L1 mRNA levels and protein levels (Fig 4D). In addition, treatment with IFN-α or IFN–γ induced abrupt increases of PD-L1 levels in BALB/c mice, which was observably suppressed by fludarabine (Fig 4E). Based on these results, it can be demonstrated that IFN-induced PD-L1 expression is mainly via upregulation and activation of its transcription factor Stat1.

### Blockage of PD-L1/PD-1 interaction enhances the HBV-specific T cell response and facilitates viral clearance in HBV transgenic mice

Finally, based on our previous studies showing that a heat shock protein gp96-based therapeutic vaccine induces a potent antiviral T cell response in HBV transgenic mice [24, 25], we further investigated possible synergy between the therapeutic vaccine and PD-L1/PD-1 blockage

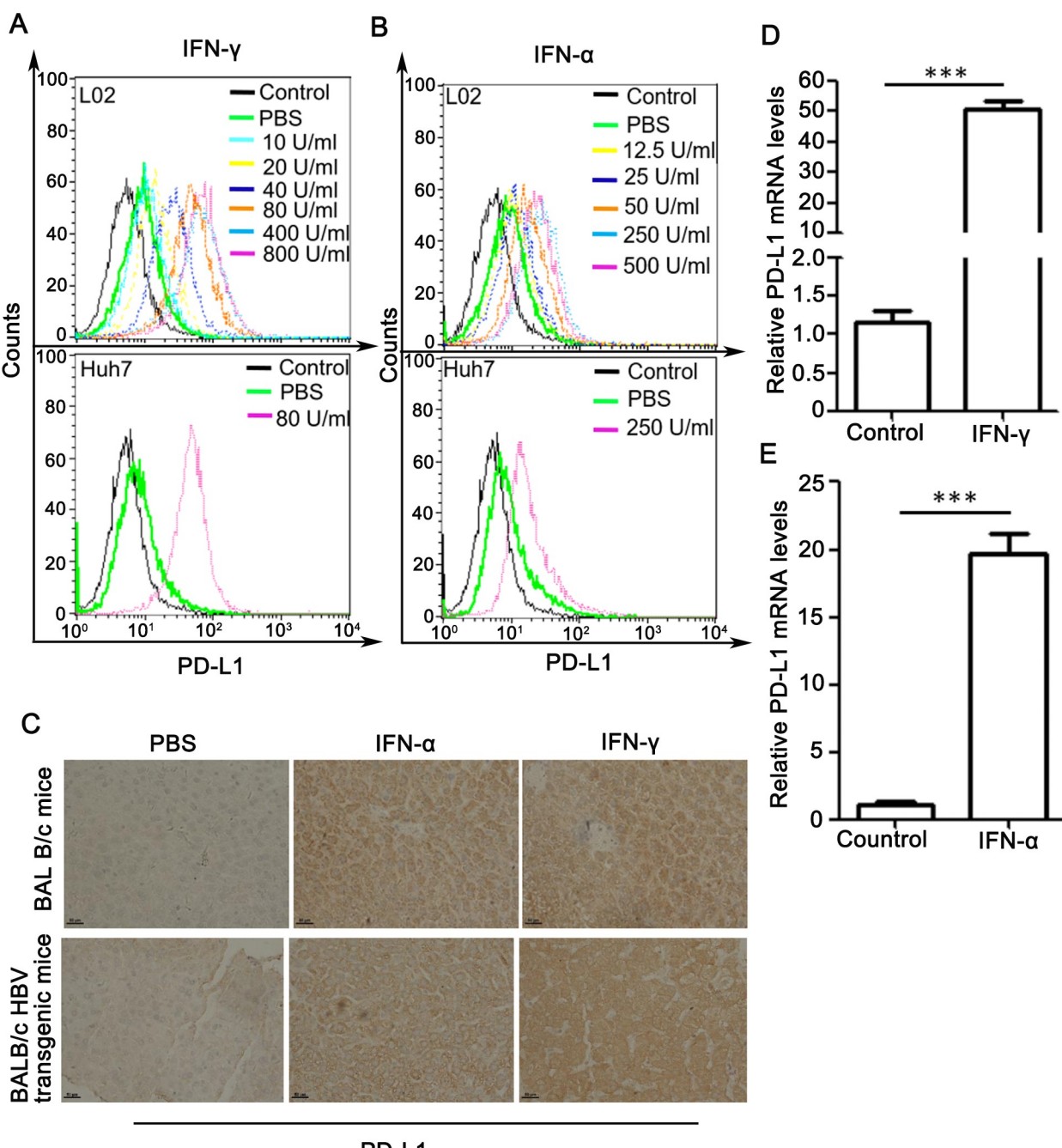

**Fig 1. Effect of IFN-γ and IFN-α on PD-L1 expression both in vitro and in vivo.** (A and B) L02 and Huh7 cells were treated with IFN-γ (A) or IFN-α (B) at indicated concentrations for 48 h. Flow cytometric analysis was performed to detect cell membrane PD-L1 levels. Cells stained with control IgG served as a negative control. (C) BALB/c or BALB/c HBV transgenic mice were treated with PBS, IFN-α, IFN–γ, as described in Materials and Methods. IHC analysis was performed for detection of PD-L1 expression levels in mouse livers. Scale bars, 50 μm. (D and E) Real-time PCR analysis of PD-L1 mRNA levels in L02 cells treated with IFN-γ (D) or IFN-α (E) or PBS as control for 48 h. The relative mRNA levels of PD-L1 were normalized to the housekeeping gene actin. Data are presented as mean ± SD for three independent experiments. *** $p < 0.001$ compared to the control.

on induction of anti-HBV T cell immunity. HBV transgenic mice were vaccinated with gp96 vaccine containing gp96 adjuvant, HBsAg and HBcAg, along with four doses treatment with

**Fig 2. Identification of Stat1 as a key transcription factor of PD-L1 induced by IFN—α/γ.** (A) Sequence analysis of the PD-L1 promoter was performed online at http://gpminer.mbc.nctu.edu.tw/. The putative binding sites for several cis-acting elements are indicated. The mutations

within Stat1 binding site were shown by italic letters. (B and C) Real-time PCR analysis of mRNA levels of putative transcription factors in L02 cells treated with IFN-α (B) or IFN-γ(C) or PBS as control for 24 h. The mRNA levels of control were arbitrarily set as 1.0. (D) L02 cells treated with 80 U/ml IFN-γ or 50 U/ml IFN-α or PBS as control for 48h and Stat1 and phosphorylated Stat1 (p-Stat1) levels were determined by western blot. (E) L02 cells were co-transfected with PD-L1 promoter luciferase reporter plasmid with a wild type (PD-L1 wt) or mutated Stat1 binding site (PD-L1 mut) and pCMV-Stat1 (Stat1) or pCMV as a mock. The relative luciferase activity was determined using dual luciferase assay kit 48 h later. (F) Cell membrane PD-L1 levels in L02 cells transfected with pCMV-Stat1 (Stat1) or pCMV as a mock were determined by flow cytometry. (G) L02 cells were co-transfected with PD-L1 promoter luciferase reporter plasmid (PD-L1 wt) and different doses of pCMV-Stat1 (Stat1) or pCMV as a mock. The relative luciferase activity was determined using dual luciferase assay kit 48 h posttransfection. Data are presented as mean ± SD for three independent experiments. *** $p<0.001$ compared to the control.

an anti-PD-1 mAb. As shown in Fig 5B, gp96-based therapeutic vaccine induced IFN-γ-secreting CD8$^+$ and CD4$^+$ T cells in mouse livers, and importantly, co-treatment with anti-PD-1 mAb resulted in a significant increase in IFN-γ-secreting T cells compared to the vaccine alone(all P < 0.05 or 0.01). Similar results were obtained in ELISPOT assay (Fig 5C). Besides, significant increases of antiviral T cell responses in the spleen of immunized mice were also observed under treatment with anti-PD-1 mAb (Fig 5D and 5E).

Significant decreases in serum HBV DNA and HBsAg levels were observed in gp96 vaccine-immunized mice treated with anti-PD-1 mAb compared to untreated mice (both P < 0.05) (Fig 6A and 6B). Mice receiving the combined treatment of gp96 vaccine with anti-PD-1 exhibited significantly reduced hepatic HBcAg expression compared to either treatment alone (anti-PD-1 vs anti-PD-1+gp96 vaccine, 53±7.6 vs 4±1, P < 0.001; gp96 vaccine vs anti-PD-1+gp96 vaccine, 20±3 vs 4±1, P < 0.001) (Fig 6C). Moderate elevation of the serum ALT levels was observed in anti-PD-1 mAb-treated and gp96 vaccine-immunized mice (Fig 6D). Taken together, these data indicate that blockage of the interaction between PD-1 and PD-L1 which expression may be upregulated by IFN-γ-secreting CD8$^+$ and CD4$^+$ T cells significantly enhances vaccine-mediated T cell response against HBV.

## Discussion

In this study, our findings identified Stat1 as the key transcription factor for hepatic PD-L1 expression through its phosphorylation induced by IFN-α/γ. Importantly, the blockage of PD-L1/PD-1 enhanced vaccine-induced antiviral T cell responses in HBV transgenic mice. Therefore, we present a new model in which IFN-α/γ activates Stat1 which promotes PD-L1 transcription and expression, and elevated PD-L1 expression in liver may contribute to impaired T cell responses, hampering the development of the virus-specific immune response in CHB.

Previous studies show that PD-L1 expression is regulated by the transcription factor Stat3 in T cell lymphoma and NF-κB in myelodysplastic syndromes blasts [28, 29]. Besides, in melanoma cells treated with IFN-γ, the activated IRF1 binds to the promoter of PD-L1 to regulate its PD-L1 [30–32]. In screening of potential transcription factors of PD-L1 in hepatocytes we found that Stat1 induced the most significant upregulation of PD-L1, suggesting that the regulation of PD-L1 expression can be exquisitely cell-type specific. In this study, IFN-α/γ pronouncedly promoted hepatic Stat1 phosphorylation both *in vitro* and in mice, likely through the Jak/Stat1 pathway [33, 34]. Furthermore, inhibition of Stat1 activation by its inhibitor or Stat1 depletion by RNAi largely abolished IFN-α/γ -mediated PD-L1 upregulation. Our results therefore indicate that Stat1 acts as a major transcription factor in IFN-α/γ -induced PD-L1 elevation in liver. Besides Stat1, we observed an increase of IRF1 expression in IFN-α/γ-treated hepatocytes (Fig 2B), and IFN-α/γ could moderately enhance the activity of PD-L1 luciferase reporter with the mutated Stat1 binding site (Fig 3A and 3C). The results indicate that IRF1 may play a minor role in IFN-α/γ -mediated PD-L1 upregulation.

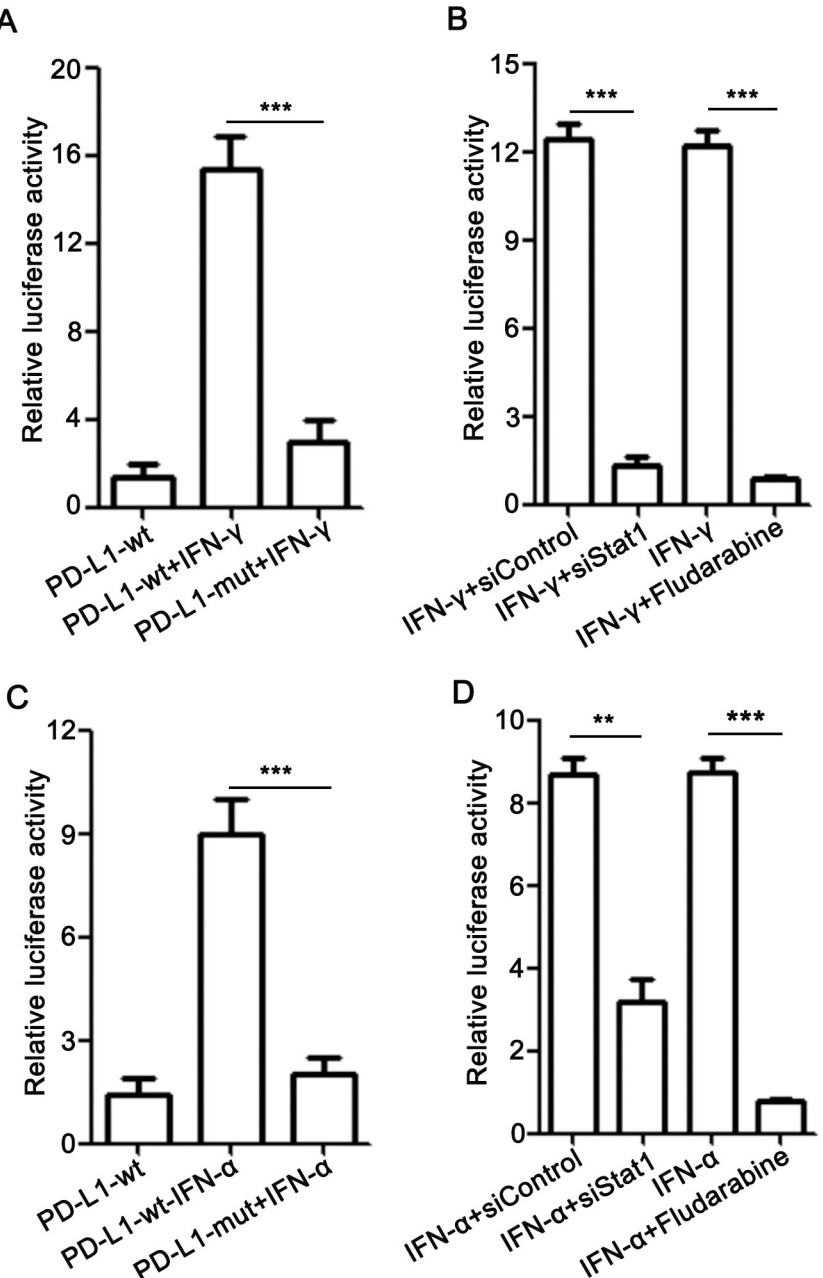

**Fig 3. IFN-α/γ upregulate PD-L1 expression in a Stat1 dependent manner.** (A and C) L02 cells were transfected with PD-L1 promoter luciferase reporter plasmid with a wild type or mutated Stat1 binding site and incubated with 80 U/ml IFN-γ (A) or 50 U/ml IFN-α (C) for 48 h. (B and D) L02 cells transfected with PD-L1 promoter luciferase reporter plasmid were co-treated with IFN-γ (B) or IFN-α (D) and Stat1 siRNA or fludarabine (5 μg/ml) for 48 h. The relative luciferase activity was determined using dual luciferase assay. Data are presented as mean ± SD for three independent experiments. $^{**}$ $p<0.01$, and $^{***}$ $p<0.001$ compared to the control.

IFN-γ and TNF-α, produced by T cells, reduce levels of HBV cccDNA in hepatocytes by inducing deamination and subsequent cccDNA decay [35]. However clinical evidence shows that the CD8+ T cells in CHB patients lose their antiviral function and ability to proliferate, which is characterized by T cell exhaustion, suppressed cytokine production and excessive inhibitory signals [36–38]. The co-inhibitory receptor PD-1, expressed on T-cells, delivers

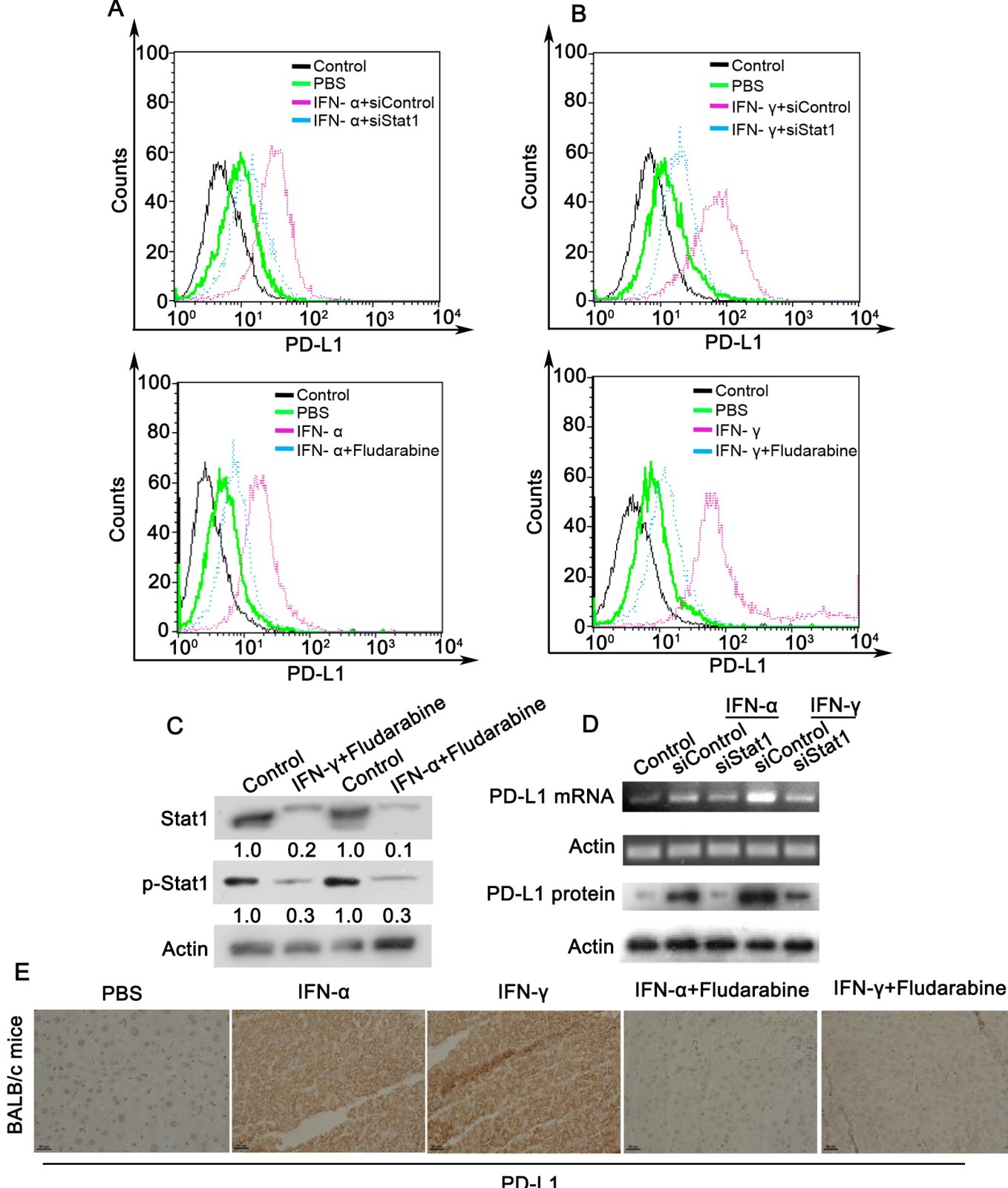

**Fig 4. Inhibition of Stat1 abrogates IFN-α/γ-induced upregulation of PD-L1.** (A and B) Cell membrane PD-L1 levels were detected by flow cytometry in L02 cells co-treated with 80 U/ml IFN-α (A) or 50 U/ml IFN-γ (B), and Stat1 siRNA or fludarabine (5 μg/ml) for 48 h. Cells stained with control IgG served as a

negative control. (C and D) L02 cells co-treated with 80 U/ml IFN-α or 50 U/ml IFN-γ, with or without fludarabine (5 μg/ml) or Stat1 siRNA for 48 h. Stat1 and phosphorylated Stat1 (p-Stat1) in cells co-treated with or without (control) fludarabine were determined by western blot (C). The mRNA and protein levels of PD-L1 were analyzed using real-time PCR and western blotting, respectively (D). (E) BALB/c mice were treated with PBS, IFN-α, and IFN–γ, and fludarabine as described in Materials and Methods. IHC analysis was performed for detection of PD-L1 expression levels in mouse livers. Scale bars, 50 μm. The experiments were performed twice with similar results.

negative signals when engaged by its ligand PD-L1, expressed on dendritic cells, macrophages, endothelial cells and hepatocytes; to attenuate T cell activation, effector functions and survival [39]. Recent studies show that PD-1/PD-L1 pathway contributes to the suppression of HBV-specific T cell function in both HBV transgenic mice and CHB patient [4, 40–43]. In addition, treatment with anti-PD-1 or anti-PD-L1 mAbs results in the enhancement or restoration of antiviral T cell function in mice [40, 41, 44–46]. Furthermore, blockage of the interaction between PD-1 and PD-L1 by specific antibodies also leads to restoration of HBV-specific T cell function, enhanced antiviral immunity, and HBsAg decline in CHB [44, 45, 47–49]. In this study, we found that the IFN-α/γ -Stat1 axis may play a role and serve as a potential drug target for hepatic PD-L1 expression which may lead to T cell inactivation, and blockage of PD-1/PD-L1 reversed liver-infiltrating virus-specific T cell activity and enhanced gp96 vaccine-induced antiviral efficiency in HBV transgenic mice.

In this study we found that IFN-γ pronouncedly promoted hepatic PD-L1 expression so it is conceivable that in CHB hepatocytes may predispose intrinsic defects in hyporesponsiveness of virus-specific IFN-γ-secreting CD8+ and CD4+ T cells by upregulation of PD-L1 expression in liver. PD-L1 overexpression may protect HBV-infected hepatocytes from T cell-mediated viral inhibition, which underlies the need for a combination strategy to directly activate T cells by vaccination and block PD-L1/PD-1-mediated immune suppression for optimal anti-HBV immunity.

Meanwhile, it is possible that PD-L1 may exert beneficial effects by preventing overactivation of inflammation and T cell responses in CHB [50, 51]. More studies are needed to dissect the immunoregulatory mechanisms of PD-L1 in various States of HBV infection, and its use as a potential prognostic marker in disease progression as seen in cancer patients [7].

IFN-α, the first drug licensed to treat HBV infection, has been extensively used as one of the major standard treatments for CHB [14, 15, 52, 53]. IFN-α promotes T cell response by stimulating cell expansion, differentiation and cytolytic function. In addition, it also has a direct antiviral effect by accelerating decay of viral capsids and inducing antiviral IFN-stimulated genes (ISGs) [54]. In this study, we found that IFN-α significantly enhances PD-L1 expression in mouse livers, which may negatively affect the anti-HBV efficiency of IFN-α. Our current work may provide further dissection of the limited effectiveness of therapeutic IFN-α in CHB and beneficial help in the design of a more efficient combined anti-HBV therapy for this first-line drug.

In conclusion, our study provides further understanding of IFN-α/γ-induced PD-L1 function in the complex regulatory networks that orchestrate T cell immune defects in chronic viral infections. Given the broad immunoinhibitory function of PD-L1 and current promising anti-PD-L1/PD-1 therapies in cancer, our work provides valuable insights into IFN-induced PD-L1 elevation in hepatic microenvironment immunotolerance and raises further potential interest in enhancing the anti-HBV efficacy of therapeutic HBV vaccines and IFN-α by blocking PD-L1/PD-1.

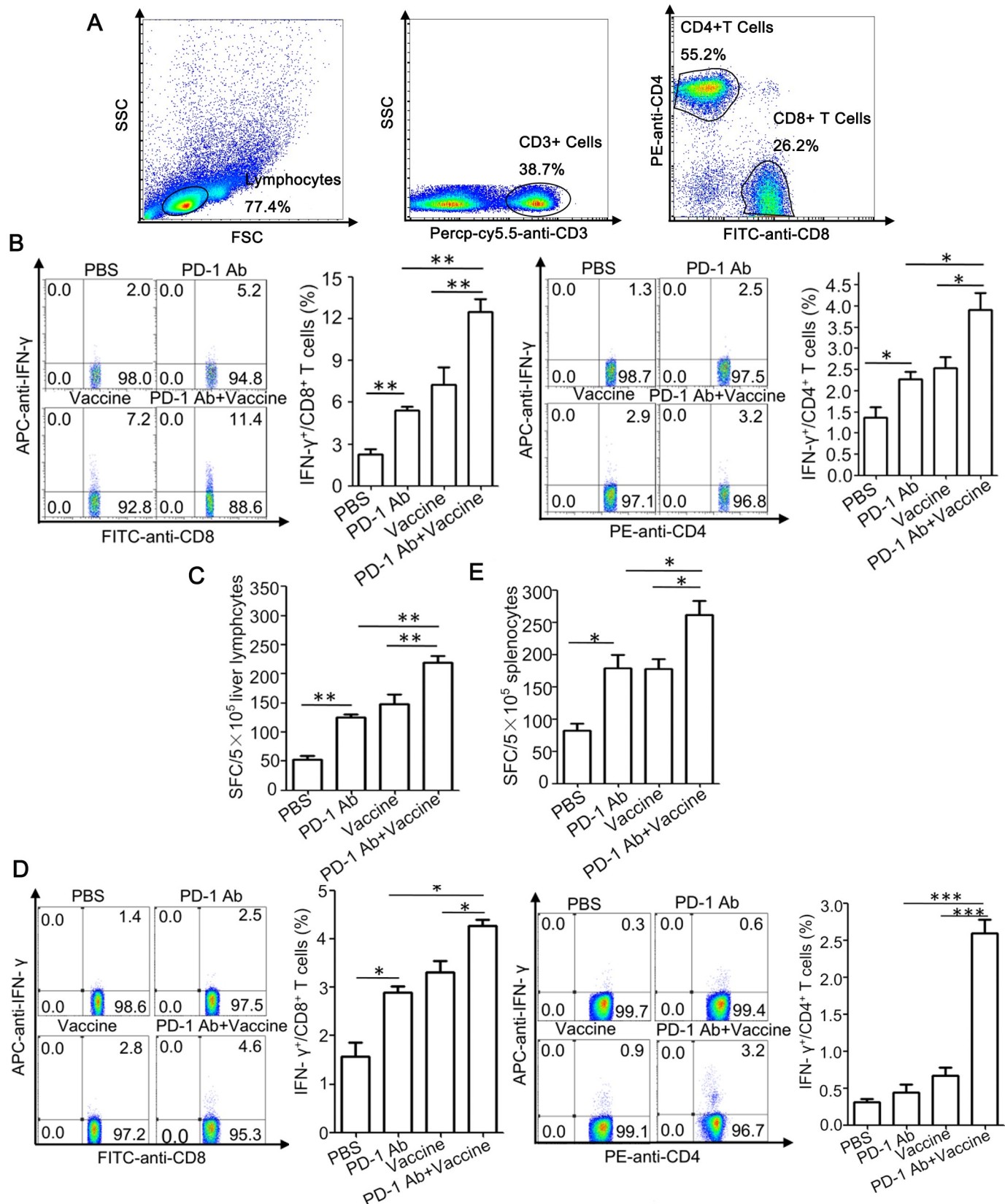

**Fig 5. Anti-PD-1 mAb treatment enhances the HBV-specific T cell responses induced by gp96 therapeutic vaccine in HBV transgenic mice.** HBV transgenic mice were immunized with the therapeutic vaccine containing HBsAg, HBcAg, and gp96 adjuvant at wks 1, 2, and 4, and/or treated with anti-PD-1 mAb (n = 5 mice/group). Mice were sacrificed at wk 9 for immunological analysis. The gating strategy for CD4 + and CD8 + T cells in lymph nodes and splenic lymphocytes by flow cytometry(A). Flow cytometric analysis was performed to quantify IFNγ+CD8+ or IFNγ+CD4+ T cell populations in the liver (B) or spleen (D) of mice. For IFN-γ ELISPOT assay, lymphocytes from liver (C) or spleen (E) (5 × 10^5 cells/well) were stimulated with HBsAg/HBcAg (5 μg/ml each), or BSA as a negative control for background evaluation. Data are presented as the mean ± SD for five mice from two independent experiments. *P < 0.05, **P < 0.01, and ***P < 0.001 using t-tests.

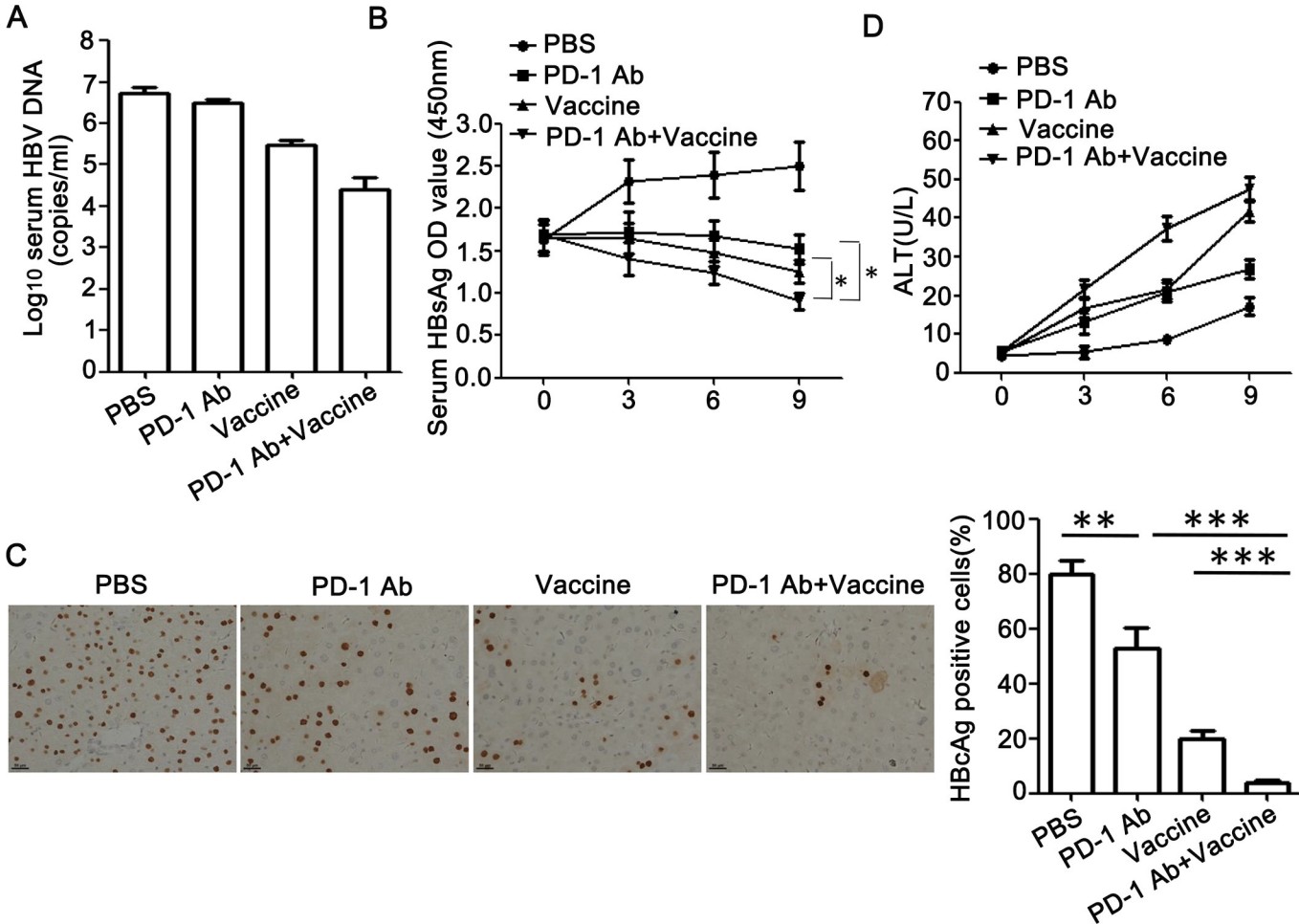

**Fig 6. Suppression of HBV expression and replication by gp96 the therapeutic vaccine is enhanced by treatment with anti-PD-1 mAb.** HBV transgenic mice were immunized and treated as in Fig 5. Mice were sacrificed at wk 9 for virological analysis. (A) Serum HBV DNA levels were quantified by real-time PCR. (B) Serum HBsAg was detected by ELISA at wks 0, 3, 6, 9, respectively. (C) IHC analysis of HBcAg expression in mouse liver tissues. The HBcAg-positive hepatocytes were counted in ten random fields under the microscope and the average number was calculated. Scale bars, 50 μm. (D) Serum ALT levels were detected by ELISA at wks 0, 3, 6, 9. Data are presented as the mean ± SD for five mice from two independent experiments. *P < 0.05, **P < 0.01, and ***P < 0.001 using t-tests.

## Supporting information

**S1 Fig. Raw images for Fig 1C.**
(PDF)

**S2 Fig. Raw images for Fig 2D.**
(PDF)

**S3 Fig. Raw images for Fig 4C, 4D and 4E.**
(PDF)

## Acknowledgments

The authors are grateful to Beijing Combio Company for the anti-PD-1 monoclonal antibody (mAb). and thankful to Beijing Tiantan Biological Products Company for the HBs protein.

## Author Contributions

**Conceptualization:** Songdong Meng.

**Formal analysis:** LanLan Liu, Junwei Hou.

**Funding acquisition:** Songdong Meng.

**Investigation:** LanLan Liu, Junwei Hou, Yuxiu Xu, Lijuan Qin, Weiwei Liu, Han Zhang, Yang Li, Mi Chen, Mengmeng Deng, Bao Zhao, Jun Hu, Huaguo Zheng.

**Supervision:** Songdong Meng.

**Writing – original draft:** LanLan Liu, Junwei Hou.

**Writing – review & editing:** Changfei Li, Songdong Meng.

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
