## [Decision Letter · Decision Letter 0]

11 Feb 2020

PONE-D-20-00809

PD-L1 upregulation by IFN-α/γ-mediated Stat1 suppresses anti-HBV T cell response

PLOS ONE

Dear Dr Meng,

Thank you for submitting your manuscript to PLOS ONE. After careful consideration, we feel that it has merit but does not fully meet PLOS ONE’s publication criteria as it currently stands. Therefore, we invite you to submit a revised version of the manuscript that addresses the points raised during the review process concerning analysis of the results, methodology, figure and legends improvments.

We would appreciate receiving your revised manuscript by 2 months. To enhance the reproducibility of your results, we recommend that if applicable you deposit your laboratory protocols in protocols.io, where a protocol can be assigned its own identifier (DOI) such that it can be cited independently in the future. For instructions see: http://journals.plos.org/plosone/s/submission-guidelines#loc-laboratory-protocols

We look forward to receiving your revised manuscript.

Kind regards,

Isabelle Chemin, PhD

Academic Editor

PLOS ONE

Journal Requirements:

Reviewers' comments:

Reviewer's Responses to Questions

**Comments to the Author**

1. Is the manuscript technically sound, and do the data support the conclusions?

Reviewer #1: Partly

2. Has the statistical analysis been performed appropriately and rigorously? 

Reviewer #1: Yes

3. Have the authors made all data underlying the findings in their manuscript fully available?

Reviewer #1: No

4. Is the manuscript presented in an intelligible fashion and written in standard English?

Reviewer #1: Yes

5. Review Comments to the Author

Reviewer #1: The manuscript by Meng et al describes a mechanism by which IFNa/g induces pSTAT-1 which then activates directly (or indirectly) PDL1 expression in vitro and in vivo liver cells. They go on to show that in HBV transgenic mice, PDL1/PD1 blockage along with HBs and HBc vaccines permit CD8 and CD4 T cells to produce IFNg.

They address an important question about the use of IFNa as a treatment, in CHB patients, may fail due to the induction of PDL1 which consequently leads to a loss of CD8 T cells responses.

Minor

The paper is well written yet requires some English improvements and the figure quality should be improved.

Major

I have several problems with the experiments that try to address the first and latter conclusions.

Figure 1: Cell counts on FACS the axis should be shown, do all the cells express PDL1 or just a percentage? mRNA levels are normalised to which house-keeping gene?

Figure 2: IRF1 has been ignored, IRF1 does have an effect on PDL1 expression. This should be commented on.

Comment: No dose experiments with pSTAT 1 construct has been shown with the PDL1 promoter.

Comment: What is the mSTAT mutation on the PDL1 promoter, a figure should be shown, or quoted. How was it predicted?

The authors should show that inhibitor fludarabine is able to block endogenous p stat induced by IFNa/g in liver cells.

Figure 4: No controls for siRNA STAT1 have been shown here.

Figure 5: no gating strategy for CD8 T cells? Neither for CD4 T cells. This is important to show.

Comment: IFNa/g treatment in the transgenic HBV mice would have increased PDL1 and drive dysfunctional T cells in terms of less IFNg production? Was this tested? Were increased HBsAg, HBeAg , ALT etc observed in the serum?

Comment: T cell tolerance, this is acquired through T cell development, T cell exhaustion is a chronic stimulation.

6. PLOS authors have the option to publish the peer review history of their article (what does this mean?). If published, this will include your full peer review and any attached files.

Reviewer #1: No

---

## [Author Response · Author response to Decision Letter 0]

30 Apr 2020

Dear Editor/Reviewers：

Thanks a lot for your attention to our manuscript. We have studied the valuable comments carefully, and tried our best to revise the manuscript. The point by point responds to the reviewer’s comments are listed as following:

Reviewer #1 (Comments for the Author):

The manuscript by Meng et al describes a mechanism by which IFN-α/γ induces pStat-1 which then activates directly (or indirectly) PD-L1 expression in vitro and in vivo liver cells. They go on to show that in HBV transgenic mice, PD-L1/PD1 blockage along with HBs and HBc vaccines permit CD8 and CD4 T cells to produce IFN-γ.

1. Figure 1: Cell counts on FACS the axis should be shown, do all the cells express PDL1 or just a percentage? mRNA levels are normalised to which house-keeping gene?

Answer: According to your suggestion, cell counts have been added to the Y axis and the intensity of fluorescence was shown in the X axis (Please see Fig 1A and 1B, Fig 2F, Fig 4A and 4B). All the cells express PD-L1, but the expression levels of PD-L1 on the cells are obviously different. The expression levels of PD-L1 on the cell surface were significantly increased after stimulation with different concentrations of IFN-α or IFN-γ. The relative mRNA levels of PD-L1 in Figure 1D are normalized to the housekeeping gene actin. The following sentence has been added in the figure legend (please see lines 177-178). “The relative mRNA levels of PD-L1 are normalized to the housekeeping gene actin.” 

Thanks for your comments which greatly strengthened this paper.

2. Figure 2: IRF1 has been ignored, IRF1 does have an effect on PD-L1 expression. This should be commented on.

Answer: The sentence “IFN-α/γ induce PD-L1 expression through activation of Stat1.” has been changed to “IFN-α/γ induce PD-L1 expression mainly through activation of Stat1.” to suggest that IRF1 also has an effect on PD-L1 expression (Please see lines 181-182). According to your suggestion, the following sentences have been added in the Discussion section (Please see lines 324-325 and 333-337) “Besides, in melanoma cells treated with IFN-γ, the activated IRF1 binds to the promoter of PD-L1 to regulate its PD-L1[30-32].”; “Besides Stat1, we observed an increase of IRF1 expression in IFN-α/γ-treated hepatocytes ( Fig 2B), and IFN-α/γ could moderately enhance the activity of PD-L1 luciferase reporter with the mutated Stat1 binding site ( Fig 3A and 3C). The results indicate that IRF1 may play a minor role in IFN-α/γ -mediated PD-L1 upregulation.”

Thank you.

3. Comment: No dose experiments with pStat 1 construct has been shown with the PD-L1 promoter. 

Answer: As you suggested, L02 cells were co-transfected with PD-L1 promoter luciferase reporter plasmid (PD-L1-wt) and different doses of pCMV-Stat1 (Stat1) or pCMV as a mock. The results showed that compared to mock, PD-L1 promoter activity was increased by Stat1 in a dose dependent way (Fig. 2G). The following sentences have been added, please see lines 196-197 and 213-216 “PD-L1 promoter activity was increased by Stat1 in a dose dependent manner (Fig 2G).”; “(G) L02 cells were co-transfected with PD-L1 promoter luciferase reporter plasmid (PD-L1 wt) and different doses of pCMV-Stat1 (Stat1) or pCMV as a mock. The relative luciferase activity was determined using dual luciferase assay kit 48 h posttransfection.”.

Thank you.

4. Comment: What is the mStat mutation on the PDL1 promoter, a figure should be shown, or quoted. How was it predicted? 

Answer: The mutant Stat1 binding sequence in PD-L1 promoter was ACTGC and has been shown in Figure 2A by italic letters. The Stat1 binding site was predicted by using online tools (http://gpminer.mbc.nctu.edu.tw/). The following sentences have been added (please see lines 191-192 and 203-204). “We mutated the core site CTGAT of Stat1 binding site on the PD-L1 promoter to ACTGC and named it as PD-L1-mut (Fig 2A).”; “The mutations within Stat1 binding site were shown by italic letters.” 

Thank you.

5. The authors should show that inhibitor fludarabine is able to block endogenous pStat induced by IFN-α/γ in liver cells. 

Answer: According to your suggestion, the effect of inhibitor fludarabine on the endogenous p-stat1 induced by IFN-α/γ in L02 cells was detected by western blot. Please see Figure 4C. The following sentences have been added (please see lines 238-239 and 250-253). “Inhibition of Stat1 and p-Stat1 by fludarabine was confirmed by western blot (Fig 4C).”;(C and D) L02 cells co-treated with 80 U/ml IFN-α or 50 U/ml IFN-γ, with or without fludarabine (5 μg/ml) or Stat1 siRNA for 48 h. Stat1 and phosphorylated Stat1 (p-Stat1) in cells co-treated with or without (control) fludarabine were determined by western blot (C).”.

Thank you for your comments which greatly strengthened this paper.

6. Figure 4: No controls for siRNA STAT1 have been shown here.

Answer: As you suggested, we have added the si-Control sequence in Table 1.

Thank you for your comments.

7. Figure 5: no gating strategy for CD8 T cells? Neither for CD4 T cells. This is important to show.

Answer: We have shown the gating strategy for CD8+ T cells or CD4+ T cells. Please see Figure 5A. The following sentences have been added (please see lines 281-283). “The gating strategy for CD4 + and CD8 + T cells in lymph nodes and splenic lymphocytes by flow cytometry(A).”

Thank you for your comments.

8. Comment: IFN-α/γ treatment in the transgenic HBV mice would have increased PD-L1 and drive dysfunctional T cells in terms of less IFN-γ production? Was this tested? Were increased HBsAg, HBeAg, ALT etc observed in the serum?

Answer: As you suggested, we treated HBV transgenic mice with IFN-α/γ and found that the expression levels of PD-L1 in liver was also significantly increased (Fig. 1C). The following sentences have been added or revised (please see lines 159-163 and 173-174). “In vivo experiment, BALB/c or BALB/c HBV transgenic mice were intraperitoneally injected with IFN-α or IFN-γ, and PD-L1 expression in liver tissues were examined by IHC. Treatment with IFN-α or IFN–γ induced abrupt increases of PD-L1 levels in both HBV transgenic mice and BALB/c mice (Fig 1C).”; “(C) BALB/c or BALB/c HBV transgenic mice were treated with PBS, IFN-α, IFN–γ, as described in Materials and Methods.”. 

IFN-α/γ may promote T cell response by stimulating cell expansion, differentiation and cytolytic function. Meanwhile, IFN-α/γ could also induce PD-L1 expression by immune cells and hepatocytes as shown in this study, which may in turn inhibit T cell response. In addition, IFN-α directly inhibits HBV replication. So, it is difficult to dissect the promotive and suppressive effect of IFN-α/γ on T cells and viral replication in HBV transgenic mice. To address this issue, the following sentences have been added in the Discussion section (Please see lines 370-373). “IFN-α promotes T cell response by stimulating cell expansion, differentiation and cytolytic function. In addition, it also has a direct antiviral effect by accelerating decay of viral capsids and inducing antiviral IFN-stimulated genes (ISGs) [54].”

Thank you for your comments which greatly strengthened this paper.

9. Comment: T cell tolerance, this is acquired through T cell development, T cell exhaustion is a chronic stimulation

Answer: As you suggested, we have changed the T cell tolerance to T cell exhaustion in the manuscript (please see lines 45-46, 63, 72 and 77).

Thank you for your comments.

---

## [Decision Letter · Decision Letter 1]

9 Jun 2020

PD-L1 upregulation by IFN-α/γ-mediated Stat1 suppresses anti-HBV T cell response

PONE-D-20-00809R1

Dear Dr. Meng,

We’re pleased to inform you that your manuscript has been judged scientifically suitable for publication and will be formally accepted for publication once it meets all outstanding technical requirements.

Kind regards,

Isabelle Chemin, PhD

Academic Editor

PLOS ONE

Additional Editor Comments (optional):

Reviewers' comments:

Reviewer's Responses to Questions

**Comments to the Author**

1. If the authors have adequately addressed your comments raised in a previous round of review and you feel that this manuscript is now acceptable for publication, you may indicate that here to bypass the “Comments to the Author” section, enter your conflict of interest statement in the “Confidential to Editor” section, and submit your "Accept" recommendation.

Reviewer #1: All comments have been addressed

2. Is the manuscript technically sound, and do the data support the conclusions?

Reviewer #1: Yes

3. Has the statistical analysis been performed appropriately and rigorously? 

Reviewer #1: Yes

4. Have the authors made all data underlying the findings in their manuscript fully available?

Reviewer #1: Yes

5. Is the manuscript presented in an intelligible fashion and written in standard English?

Reviewer #1: Yes

6. Review Comments to the Author

Reviewer #1: The article PD-L1 upregulation by IFN-α/γ-mediated Stat1 suppresses anti-HBV T cell response has been revised both experimentally and within the text, in all the authors have answered all concerns.

7. PLOS authors have the option to publish the peer review history of their article (what does this mean?). If published, this will include your full peer review and any attached files.

Reviewer #1: No

---

## [Editor Report · Acceptance letter]

23 Jun 2020

PONE-D-20-00809R1 

PD-L1 upregulation by IFN-α/γ-mediated Stat1 suppresses anti-HBV T cell response 

Dear Dr. Meng:

I'm pleased to inform you that your manuscript has been deemed suitable for publication in PLOS ONE. Congratulations! Your manuscript is now with our production department. 

Kind regards, 

on behalf of

Mrs Isabelle Chemin 

Academic Editor

PLOS ONE